# Dynamic Structure of Yeast Septin by Fast Fluctuation-Enhanced Structured Illumination Microscopy

**DOI:** 10.3390/microorganisms9112255

**Published:** 2021-10-29

**Authors:** Longfang Yao, Li Zhang, Liwen Chen, Xingyu Gong, Jiahui Zhong, Baoju Wang, Yiyan Fei, Lan Mi, Jiong Ma

**Affiliations:** 1Shanghai Engineering Research Center of Ultra-Precision Optical Manufacturing, Key Laboratory of Micro and Nano Photonic Structures (Ministry of Education), Department of Optical Science and Engineering, School of Information Science and Technology, Fudan University, 220 Handan Road, Shanghai 200433, China; 17110720023@fudan.edu.cn (L.Y.); 19210860081@fudan.edu.cn (L.C.); 19210720001@fudan.edu.cn (X.G.); fyy@fudan.edu.cn (Y.F.); 2Shanghai Engineering Research Center of Industrial Microorganisms, The Multiscale Research Institute of Complex Systems (MRICS), School of Life Sciences, Fudan University, 220 Handan Road, Shanghai 200433, China; 19110700077@fudan.edu.cn; 3Institute of Biomedical Engineering and Technology, Academy for Engineer and Technology, Fudan University, 220 Handan Road, Shanghai 200433, China; 19210860047@fudan.edu.cn; 4Centre for Optical and Electromagnetic Research, Guangdong Provincial Key Laboratory of Optical Information Materials and Technology, South China Academy of Advanced Optoelectronics, South China Normal University, Guangzhou 510006, China; baoju.wang@m.scnu.edu.cn

**Keywords:** CDC12, septin, SRRF, SIM, super-resolution microscopy, living-cell image, *Saccharomyces cerevisiae*

## Abstract

When *Saccharomyces cerevisiae* divides, a structure composed of different septin proteins arranged according to a certain rule is formed at the cell division site. The structure undergoes multiple remodeling stages during the cell cycle, thus guiding the yeast cells to complete the entire division process. Although the higher-order structure of septins can be determined using electron microscopy, the septin’s dynamic processes are poorly understood because of limitations in living cell super-resolution imaging technology. Herein, we describe a high lateral resolution and temporal resolution technique, known as fast fluctuation-enhanced structured illumination microscopy (fFE-SIM), which more than doubles the SIM resolution at a frame rate of 38 Hz in living cells. This allows a highly dynamic and sparse septin structure to be observed in *Saccharomyces cerevisiae*.

## 1. Introduction

Septins exist in most eukaryotes and are conserved GTP-binding proteins that were first discovered to play a role in the cell cycle of budding yeast, S. cerevisiae [1,2]. Septins assemble into basic units; these 17–32 nm rod-shaped non-polar hetero-oligomers are polymerized end-to-end into filaments. Filaments can be organized into various higher-order structures, such as linear arrays, filaments, rings and gauzes [3]. Budding yeast is still a valuable model for the study of higher-order structures of septins, and its study may also apply to animal cells, helping to understand septin-related diseases. Seven genes have been identified in yeast, among which the four proteins Cdc3, Cdc10, Cdc11 and Cdc12 constitute the main part of the septin ring [4,5]. When yeast cells divide, septins first form a septin ring at the budding site. After the bud appears, the septin complex at the bud neck expands into an hourglass shape. Before cytokinesis, the septin ring complex splits into two rings. The process by which the septin filament controls the separation of the cell membrane between the mother and daughter cells is still not fully known [6,7].

The research methods for studying septin structures are mainly in vitro imaging technology and electron microscopy (EM). In vitro imaging is used to obtain a segment or bundle of septin complexes by protein purification and recombination, then uses super-resolution or EM imaging to deduce its structure according to the results [8,9]. Although in vitro imaging has high accuracy and resolution, the reconstructed structures in vitro cannot completely represent the natural structure in vivo. The in vivo structure of septin is mainly imaged by EM through the septin ring of division cells [10]. In EM imaging, the sample structure may change during preparation. In complex organisms, distinguishing different substances and structures is generally based on structural characteristics, and distinguishing unstable structures is difficult [11,12]. Moreover, in vitro recombination imaging and EM imaging do not perfectly reveal the transformation process in the higher-order structures of septins during cell division. Only living-cell imaging can reveal a dynamic process. The molecular rearrangement of highly ordered septin proteins during yeast cell division was observed by fluorescence polarization microscopy (FPM), accompanied by a 90° rotation of the septin filaments. These filaments are arranged parallel to the mother–bud axis in the ring, and are orthogonal to it in the double-ring state [13,14,15,16]. In addition, photobleaching fluorescence recovery experiments show that although the septin ring is rigid, the splitting septin ring is dynamic [17]. However, the resolution of the FPM is relatively low and only reveals the change in the direction of the septin protein and cannot distinguish specific structural changes.

Different populations of mobile and immobile septins are present in cells. It has been proven that there is an exchange at the ring with free cytosolic septins [18,19]. Whether the organization of mobile and immobile septins in higher-order structures is different in cells is still unknown [14]. DeMay et al. speculated that the moving parts are arranged in a disorderly and random manner, whereas the ordered parts are static, but the photobleaching recovery experiment confirmed that the dynamic septin was also ordered [14]. They speculated that a layer of resident filaments extends from the membrane that functions as a template for the filaments to polymerize and depolymerize on the cytosolic surface. Therefore, even if the part is unstable, the overall integrity of the structure is maintained [14]. Nonetheless, the functions of dynamic septin remain unclear. At present, only static and ordered structures can be distinguished under an electron microscope. Therefore, high-precision and high-resolution imaging for living cells is urgently needed to observe the dynamic aspects of septins.

Traditional super-resolution technologies, such as stimulated emission depletion microscopy (STED) and single-molecule localization microscopy (SMLM), achieve very high spatial resolution [20,21,22]. Intense excitation light is required to obtain as many photons as possible for high-resolution images, yet the number of photons emitted by fluorescent protein molecules per unit time is limited; therefore, the resolution of STED live-cell imaging is relatively low. Strong excitation light also increases photobleaching and phototoxicity, which limits its application to living cell imaging. SMLM requires thousands of images for reconstruction into a super-resolution image, and thus its temporal resolution is very low. Among the existing super-resolution microscopy techniques, SIM has advantages in terms of duration, speed and multiple application scenarios for live-cell imaging [23]. Lattice SIM requires a lower illumination intensity and fewer photons than classic SIM [24]. Although lattice SIM is the most suitable super-resolution microscopy technology for imaging living cells, its spatial resolution, at only 120 nm, is far less than that of STED (~50 nm) and SMLM (~20 nm) [25].

To overcome such limited resolution, many researchers have combined different super-resolution techniques. For example, the combination of Airyscan and SRRF increases the lateral resolution to approximately 40 nm [26]. Super-resolution radial fluctuations (SRRF) are a type of super-resolution technology based on fluorescence fluctuations. It combines temporal fluctuation analysis and local microscopy, and achieves higher resolution through additional radial symmetry operation [27]. SRRF can extract information from samples with higher density intermittent fluorescence, and allows optical resolution in exchange for the time resolution required for live cell recording. It has been reported that the combination of SIM and SRRF can image dye-labeled living cells with a resolution of 32 nm by processing a 9-frame SIM algorithm image [28]. We improved the processing method to enhance the temporal resolution of living-cell imaging and named the proposed method fluctuation-enhanced structured illumination microscopy (fFE-SIM). FFE-SIM requires two traditional SIM images to achieve the spatial resolution of 15 frames of SIM images, and the temporal resolution is increased by a factor of seven.

Using the proposed fFE-SIM method to image live budding yeast, we found Cdc12-related structures in the middle portion of the splitting septin ring, and determined that these structures are highly dynamic during cytokinesis.

## 2. Materials and Methods

### 2.1. Yeast Strains

The *Saccharomyces cerevisiae* yeast strain AGY169-pRS416-ScCdc12-conGFP:GEN(4D4) was a generous gift from Juntao Gao’s lab in Tsinghua University; it has four amino acids removed from the carboxylic groups end of the septin Cdc12 and four amino acids removed from the amino groups end of the green fluorescent protein (GFP) with no linker between them.

### 2.2. Plasmid for Mammalian Cells

All primers were purchased from Sangon Biotech (Shanghai, China) and the constructs were sequenced by Sangon Biotech. After restriction digestion to generate an empty backbone, appropriate fragments were inserted into the plasmid pLVX-Puro. Plasmid pLVX-Puro-EGFP-alphaTubulin carries an N-terminal EGFP-tagged alphaTubulin under the CMV promoter.

### 2.3. Imaging Dish Preparation 

For the yeast, the confocal imaging dishes (Glass Bottom Cell Culture Dish, 060921EC01, Nest Biotechnology, Wuxi, China) were soaked in acetone at 5 °C for 30 s and washed with ddH20 more than three times. After the glasses were dried at room temperature to remove residual solvent, they were immersed in 10 mg/mL ConA (11028-71-0, Sigma-Aldrich, Darmstadt, Germany) for 10 min. After drying for 2 h, the coverslips were washed with nuclease-free water and allowed to air dry. For the mammalian cells, the confocal imaging dishes were coated with fibronectin (FC010; Sigma-Aldrich, Darmstadt, Germany).

### 2.4. Yeast Cell Preparation

The yeast cells were inoculated into 6 mL of YPD culture medium (YPD broth, A507022, Sangon Biotech, Shanghai, China) and incubated at 30 °C and 220 rpm with agitation until log phase growth. Next, yeast cells were harvested by centrifugation at 3000 rpm and suspended in fresh culture medium. The yeast suspensions were then equally distributed onto ConA-coated plates, and covered with agar sheets to prevent shift.

### 2.5. Mammalian Cell Culture

BS-C-1 cells (ATCC, CCL-26) were cultured in Modified Eagle Medium basic (11095-080, Gibco, Thermo Fisher Scientific, Waltham, MA, USA), supplemented with 10% fetal bovine serum (Gibco, Thermo Fisher Scientific, Waltham, MA, USA) containing 100 U/mL penicillin and 100 μg/mL streptomycin (15140-122, Gibco, Thermo Fisher Scientific, Waltham, MA, USA) and non-essential amino acids (NEAA, 11140-050, Gibco, Thermo Fisher Scientific, Waltham, MA, USA). Before transfection, cells were cultured in 12-well cell culture dishes overnight to 60–80% confluence. The cells were transfected with constructs for 24 h using Lipofectamine 3000 (Thermo Fisher Scientific, Waltham, MA, USA) according to the manufacturer’s protocol. After transfection, the cells were plated on the glass of a confocal imaging dish for 2 days at 37 °C with 5% CO_2_. For fixed cell imaging, cells were harvested at approximately 30% confluence and fixed with 4% paraformaldehyde (157-8, Electron Microscopy Sciences, Hatfield, PA, USA) and 0.1% glutaraldehyde for 10 min at room temperature after treatment with extraction buffer (0.1 M PIPES, 1 mM EGTA, 1 mM MgCl2, 0.2% Triton X-100) for 1 min.

### 2.6. Imaging Acquisition and Analysis

Image acquisition was performed on the Lattice SIM (ZEISS Elyra 7) with a 63× objective lens (alpha Plan-Apochromat 63×/1.46 Oil Korr M27, ZEISS, Heidenheim, Germany). A 488 nm laser for GFP and EGFP was used for excitation illumination. The SIM images acquired at high resolution were automatically SIM processed using ZEN software (ZEN 2.3sp, ZEISS, Heidenheim, Germany). The StackReg plugin of ImageJ software was used to correct for drift and align the image stack with the target image. The fFE-SIM image reconstruction was implemented using the NanoJ-SRRF plugin in ImageJ [27]. We set the ring radius at 0.5, the radiality magnification at 5 and 6 axes in the ring for SRRF reconstruction of SIM image. The image data were measured and analyzed by ImageJ and Origin.

## 3. Results

### 3.1. Lateral Resolution Verification of fFE-SIM

In traditional SIM processing, every nine of the original image frames are reconstructed into a super-resolution image and the time interval between adjacent SIM images is the time spent on these original images. To obtain a fluent dynamic movie and increase the temporal resolution, we sliding process the images. The original images O1–O15 are processed into SIM super-resolution image S1, O2–O16 are processed into S2, On-14–On into Sn-14 and so on (Figure 1A). Thus, the time interval between two adjacent SIM super-resolution images is the time interval between the two original images. Similarly, SRRF is also conducted following the sliding process and the time interval between two adjacent SRRF images is also the time interval between two original images.

To verify the lateral resolution, we fixed BS-C-1 cells transfected with EGFP-alphaTubulin and imaged them with lattice SIM to obtain a sequence of time-series images and then them processed with the SRRF algorithm after processing with the SIM algorithm. Compared with the lattice SIM image, the microtubule diameter is thinner after SRRF processing and the signal-to-noise ratio improves (Figure 1B,C). For processing with the SRRF algorithm, the larger the number of processed frames, the higher the lateral resolution and the better the image quality, but the lower the temporal resolution. To balance the lateral and temporal resolutions, we compared the effects of SRRF processing using different numbers of frames (5, 10, 15, 30, 45) (Figure 1I–M). Because of the blinking property of EGFP, the fluorescent molecules will not all display during the brief exposure. For an objective comparison, the same number of SIM images were projected into one frame (Figure 1D–H). In the intensity profiles of the same location, the SIM image has only one peak, but the fFE-SIM image has multiple peaks. Among them, the fFE-SIM images processed using 15, 30 and 45 frames have obvious double peaks. The peak-to-peak distances are 79.5, 84.0 and 79.4 nm, respectively (Figure 1P–R). Two peaks can be distinguished only when the same bifurcation on the microtubule is 179.2 nm away from the SIM images (Appendix A). In other areas, the same resolution is attained in as few as five frames (Appendix A). The luminescence of the EGFP molecules is unstable, and randomness will occur under brief exposure. To obtain a more stable resolution in most areas, the fFE-SIM images of living cells were processed using 15 frames of the SIM image. 

The best resolution of lattice SIM is 120 nm, but EGFP cannot fully reach this resolution because of its optical properties. To ensure cell activity, the excitation light power must not be too high; therefore, the SIM resolution is approximately 180 nm. Nevertheless, the resolution is doubled to approximately 80 nm after the fFE-SIM processing. 

### 3.2. Lateral Resolution Verification of fFE-SIM in Live Cell

To validate our method in live cells, live BSC-1 cells transfected with EGFP-α-tubulin were imaged with a lattice SIM microscope. After fFE-SIM processing, the lateral resolution was similar to that of the fixed cells (Figure 1 and Figure 2A–C). A total of 2250 original images of the sample were taken sequentially for a period of 59.28 s, and each image took 26.34 ms. According to the sliding processing, 2236 frames of SIM processing images were obtained. Each SIM image represents 395.2 ms of accumulated time and the interval between two adjacent frames is 26.34 ms. Fifteen SIM images are processed into one fFE-SIM image; each fFE-SIM image contains 29 frames of original image information and represents 736.9 ms. The interval between the fFE-SIM images of two adjacent frames is 26.34 ms, and the long-term imaging frame rate is 37.9 Hz. The traditional SIM processing method involves processing every 15 frames into a super-resolution SIM image, and then processing every 15 of the SIM images into an SRRF image. Consequently, a single SRRF image frame represents 225 frames of the original image. The proposed method increased the temporal resolution by more than seven times.

Although the temporal resolution is not low, when an event occurs, the performance differs on the SIM images and the fFE-SIM images. At a microtubule bifurcation in this sample (Appendix A), one of the two bifurcated lines broke (Figure 2D,E, red arrowhead). After analysis, it was found that frames S10, S11, S12 and S13 had changed (Figure 2D), which shows that the peak shifts from the right to the left in the distribution profile (Figure 2F and Appendix A). In the fFE-SIM image, the change occurred more uniformly; that is, one of the peaks gradually decreased (Figure 2G and Appendix A). The diagram shows that when the event occurs at S15, it affects 15 of the fFE-SIM images (Figure 2H).

In summary, fFE-SIM reduces the temporal resolution of the original image; however, its lateral resolution is at least 700% higher than that of the same type of processing method.

### 3.3. Long Term Living Yeast Cell Imaging of a Cdc12 Labeled Septin Ring by fFE-SIM

During the *Saccharomyces cerevisiae* cell division cycle, the structure of the septin ring also undergoes periodic changes (Figure 3A). Here, we only focus on the process whereby the septin changes from an hourglass shape into two rings and the last two rings separate (Figure 3A, ⅲ and ⅳ). We conduct long-term imaging of the yeast in which the septin split ring divides in two (Figure 3B) and a magnification of the position of the split ring (Figure 3C). We observe that the portion near the edge is brighter and the middle is relatively dim. A line is drawn along one of the rings, and the intensity profile has two obvious peaks. The region corresponding to the two peaks is the part with stronger fluorescence and its average fluorescence intensity is termed Ioutside. The plateau between the peaks is the portion in the middle of the ring with weaker fluorescence and its average fluorescence intensity is termed Iinside (Figure 3D). The movie of the septin ring (Appendix A) shows that the outside edge of the ring is relatively dense and less dynamic, whereas the middle that should have been empty has a relatively dynamic loose structure. The structure in the middle continues to increase and the ratio of Iinside to Ioutside shows an upward trend over time (Figure 3E). However, the resolution of SIM is limited, and a clear structure cannot be determined (Figure 3F). Therefore, we use the above method to process it with fFE-SIM to examine for a clear structure (Figure 3G and Appendix A). In the fFE-SIM image, clear lines appear on one side of the ring (Figure 3G, red arrow) and some lines connect to the ring. The gray value on the SRRF image correlates, but not proportionally, with the fluorescence intensity of the original image. These short lines are not clear in the SIM images (Figure 3F, red arrow), and can be easily overlooked.

In short, during the formation of the two septin rings, dynamic structures related to the Cdc12 protein are formed in the middle of the rings. 

### 3.4. A New Architectural Model of a Cdc12 Labeled Septin Ring

The results of imaging many yeast cells in processes iii and iv (Figure 3A) lead us to a new septin architecture model (Figure 4A). Septins form a relatively dense and rigid ring close to the cell membrane at the neck of the dividing bud (Figure 4A, green part), which serves as a scaffold to support this shape. In the middle of the ring, there is another structural system composed of Cdc12 proteins (Figure 4A, red part); it is very dynamic and loose to facilitate the passage of other substances (such as microtubules, actins and myosin proteins) between the mother and daughter cells. Following the transformation of the intermediate structure, we divide the transition of the septin ring from an hourglass shape into two rings into four stages (Figure 4B). In Stage I, the hourglass-shaped septin presents two arc-shaped structures on the x–y axes and a dynamic structure begins to form between the two arcs. A specific example is the first cell in Figure 4C, depicted dynamically in Appendix A. Appendix A shows a bright part in the middle, which is connected to the peripheral dense part in the y–z direction by filaments and occasionally extends filaments in the x-direction. In Stage II, the dynamic structure in the middle becomes apparent and the network structure is continuously produced in the x, y and z directions. On the x–y section of the septin ring, the two arcs begin to break in the middle, which is the second cell in Figure 4C In Stage III, the position of the dynamic structure tends to be on the same y–z plane as the outer rigid structure. The x–y section is barely discernable between two lines, but the two lines are closely connected (Figure 4C, the third cell). The middle of these two lines is changing at a high speed and the arc breaks in the middle. The cell in Figure 3 has reached this stage. In Stage IV, the two rings separate completely and the structure in the middle is gradually dissociated with little dynamic in the x-direction, mainly on the y–z plane. In this stage, the dense structure of the periphery also dissociates and the shape of the x–y section changes from an arc to dots (Figure 4C, fourth cell). Of course, the boundaries between these four periods are not apparent and it is difficult to assign cells accurately to a certain period. To validate this model, we identified three additional cells at each stage (Appendix A).

In conclusion, Cdc12 participates in the formation of the peripheral rigid scaffold and the dynamic structure in the middle of the septin ring, which first increases and then decreases.

## 4. Discussion

How the internal proteins of the septin ring change during yeast division is of great significance in studying how cells use the same group of proteins to achieve fine control of different cell processes. Different septin complexes that are located on the plasma membrane, cilia, sperm rings and dendrites are involved in many cellular processes, including cytokinesis, ciliogenesis, morphogenesis, mitosis and exocytosis [29,30]. Septin protein functions as scaffolding for molecular aggregation into a higher-order structure and regulates membrane movement by binding its AH domain with lipids [31]. However, the septin protein on the neck of the mother bud controls the separation between the mother and daughter cells. Most of the methods for investigating the architecture of the septin ring fix the cell and observe the structure of its dense and orderly part using an electron microscope. Sometimes, the cell wall is removed, severely damaging the fine structure of the cell. Moreover, an electron microscope cannot distinguish the type of protein or the composition of the septin outside the dense and orderly architecture. Living cell imaging can be used to study the process of septin transformation. The study of living cells under the FPM shows that septin filaments rotate by 90° during the transition from hourglass to double ring [13,14]. The photobleaching recovery experiment shows that the disassembly and reassembly of filaments are involved in the process of rotation [17]. However, the resolution of the FPM is inadequate for discerning a septin structure in detail. A living-cell imaging method was designed with high lateral resolution and high temporal resolution to observe the dynamics of the septin ring during cell division from hourglass to double-ring in the bud neck of *Saccharomyces cerevisiae*. Because the quantum efficiency of the fluorescent protein is not high and is easy to quench, it does not present the best temporal and spatial resolution. If we use dyes with better optical properties for imaging, our temporal resolution can reach 255 Hz.

In general, we found a dynamic and loose Cdc12 protein related structure in the middle of the septin ring, which changed with the cell cycle. This structure has previously been shown in images obtained by FPM, but failed to attract research attention [15,16]. In these images, septin was split into two rings, showing the shape of the two lines. If the ring is hollow, the two parallel rings will not appear as two lines, but as four points or four short lines under the high-power lens. The double line indicates that there is an additional structure surrounded by the ring. In the SIM movies, it is obvious that the middle structure of the septin ring is highly dynamic. In the fFE-SIM movies, the structure is composed of small linear objects that agglomerate into a loose network structure. The fFE-SIM processing results in a decrease in temporal resolution and because of the highly dynamic structure, some of the linear objects may be motion artifacts. These loose and dynamic structures in the middle of the septin ring may be to provide remodeling materials for the dense and ordered part of the septin ring, and the AH domain of Cdc12 may recruit membrane-related materials to the division site to form a new cell membrane between the mother and daughter cells [32]. It may also be that Cdc12 acts as scaffolding for the interaction of microfilaments and microtubules in the middle, allowing them to pass through the pores in the septin ring, forcing the daughter cells out of the mother cells and then breaking the microfilaments and microtubules [33]. 

In conclusion, we improved a long-term super-resolution imaging method for labeling living cells with a fluorescent protein, at a resolution of approximately 80 nm at a frame rate of 38 Hz. Using this method, we identified a new higher-order structure of the septin protein. These data will serve as a novel platform to better understand cell division and the various complex biological processes related to septin.

## Figures and Tables

**Figure 1 microorganisms-09-02255-f001:**
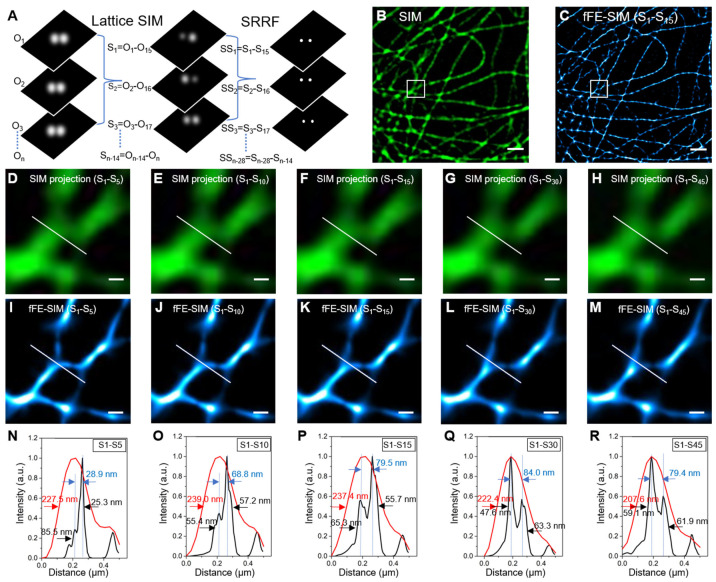
The resolution of fFE-SIM in fixed cells transfected with EGFP-a-Tubulin. (**A**) Schematic of fFE-SIM processing. Left: a series of original images. Middle: a series of reconstructed SIM images. Right: a series of reconstructed fFE-SIM images. (**B**) Image of subcellular microtubules labeled with EGFP by SIM algorithm processing (scale bar, 1 µm). (**C**) Reconstructed fFE-SIM image by 15 frames of SIM images (scale bar, 1 µm). (**D**–**H**) Sum projections by 5, 10, 15, 30 and 45 SIM images indicated by the white box in (**B**) respectively (scale bar, 200 nm). (**I**–**M**) Reconstructed fFE-SIM image by 5, 10, 15, 30 and 45 SIM images indicated by the white box in (**B**) respectively (scale bar, 200 nm). (**N**–**R**) Corresponding line-scanning profile from the SIM projection images (red line) and fFE-SIM images (black line) processing by 5, 10, 15, 30 and 45 SIM images, respectively. The red and black one-way arrows indicate FWHM. The blue arrows indicate the peak-to-peak distance.

**Figure 2 microorganisms-09-02255-f002:**
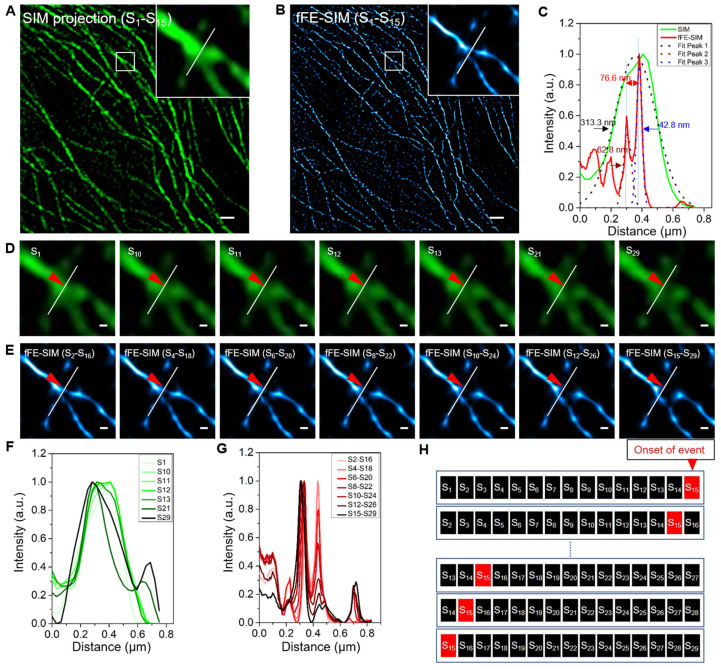
Application of fFE-SIM in microtubule imaging of living cells. (**A**) Sum projection by 15 SIM images of microtubules labeled with EGFP. The inset in upper right zooms in on the small box at center (scale bar, 1 µm). (**B**) Reconstructed fFE-SIM image by 15 frames of SIM images. The inset in upper right zooms in on the small box at center (scale bar, 1 µm). (**C**) Intensity profiles along the white lines in (**A**) and (**B**) respectively. The double-sided arrows indicate the peak-to-peak distance. Dot lines indicate the Gaussian fit used to determine the FWHM (one-way arrows). (**D**,**E**) For the time-lapse images of the white-boxed region in (**A**) and (**B**) respectively, representative frames are displayed at a magnified scale (scale bar, 100 nm). (**F**-**G**) Intensity profiles along the white lines in (**D**) and (**E**), respectively. The red-arrowheads indicate the location of the event. The analysis of the rest of the images are shown in Appendix A. (**H**) Schematic diagram of event mutation homogenized by fFE-SIM algorithm processing.

**Figure 3 microorganisms-09-02255-f003:**
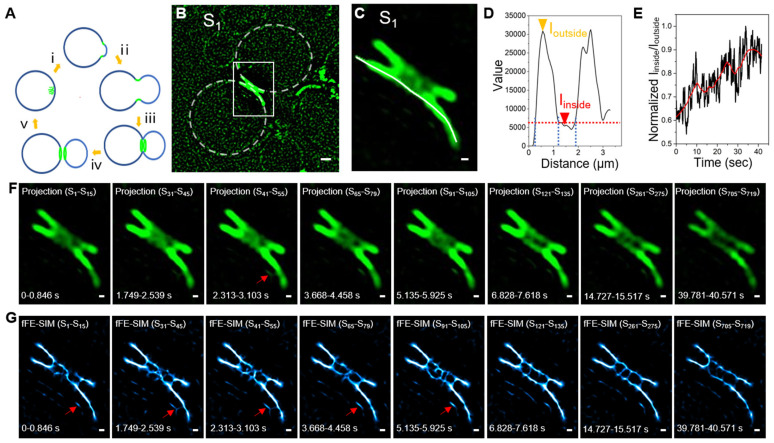
Imaging and analysis of Cdc12-GFP in living yeast with fFE-SIM. (**A**) Change pattern of septin ring in yeast cell division cycle. The green indicates the septin proteins. (**B**) SIM image of GFP-Cdc12. Dashed lines indicate the cell edges of mother cell and daughter cell (scale bar, 1 µm). (**C**) Enlarged images of the white-boxed region in (**B**) (scale bar, 200 nm). (**D**) Intensity profiles along the white lines in (**C**). The yellow-arrowhead indicates the average gray value of the brighter part around the septin ring (I_outside_). The red-arrowhead indicates the average gray value of the darker part inside the septin ring (I_inside_). (**E**) Variation curve of I_inside_ and I_outside_ ratio with time. (**F**,**G**) For the time-lapse images of SIM and fFE-SIM processing, representative frames are displayed (scale bar, 200 nm). The red-arrow indicates short lines connect to the ring.

**Figure 4 microorganisms-09-02255-f004:**
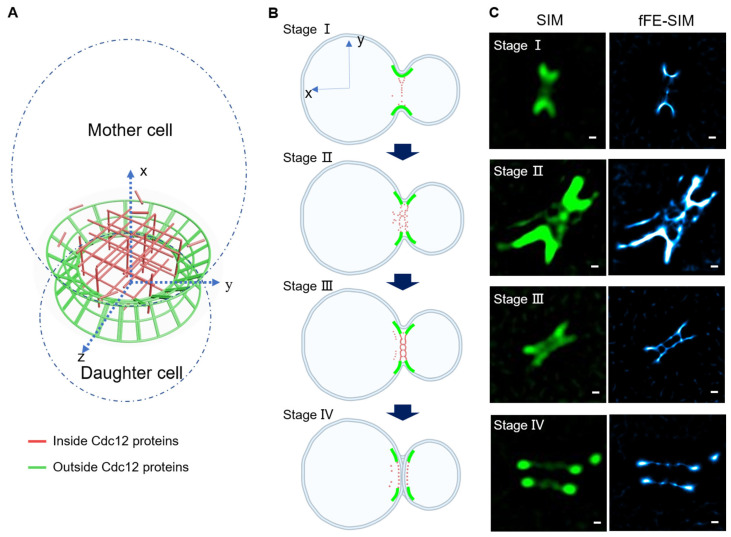
The architectural model and examples of Cdc12-GFP labeled septin ring. (**A**) 3D schematic diagram of a septin ring. Dashed lines indicate the cell edges of mother cell and daughter cell. Green indicates the Cdc12 proteins on the septin ring. Red indicates the Cdc12 proteins inside the septin ring. (**B**) Septin ring viewed in the xy of (**A**) at different stages of hourglass-to-double ring transition. (**C**) One example for each stage in (**B**). Three other examples are shown in Appendix A (scale bar, 200 nm). Left: SIM images. Right: fFE-SIM images.

## Data Availability

Not applicable.

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
