# Peer review of "Dynamic Structure of Yeast Septin by Fast Fluctuation-Enhanced Structured Illumination Microscopy"

_microorganisms, 2021, doi:10.3390/microorganisms9112255_

Round 1

Reviewer 1 Report

In manuscript entitled “Dynamic structure of yeast septin by fast fluctuation-enhanced structured illumination microscopy” authors describe visualisation of septin in yeast S. cerevisiae by improved superresolution microscopy technique referred to a fFE-SIM, which is able to visualize dymanic processes with high lateral and temporal resolutions. Using the technique they record dynamics of septin ring, which they observe as relatively stable outer ring and a highly dynamic array of septin molecules in the inner ring. I find their data interesting. Interpretation of data and discussion appears to be adequate.

I have, however few minor concerns: 

  • Lines 114-117 “it has four amino acids removed from the 3′ end of the septin Cdc12 and four amino acids removed from the 5′ end of the green fluorescent protein (GFP) with no linker between them.“ If they are writing about aminoacids, it should not be 5’ and 3’ end (which in on DNA).
  • English must be carefully edited.

Author Response

Thanks for the comment. That's a good suggestion. We'll make changes Lines 116-119 of the latest manuscript into " it has four amino acids removed from the C end of the Cdc12 and four amino acids removed from the N end of the green fluorescent protein (GFP) with no linker between them." 

Reviewer 2 Report

This report describes the performances of a microscopy method developed by the authors and combining SIM microscopy with a processing approach allowing to enhance the temporal resolution of the so called fFE-SIM method. The authors have chosen to analyze the organization of septins by tagging Cdc12 fluorescently at the bud neck of dividing yeasts, as a proof of principle. They achieve a lateral resolution of about 80 nm with an imaging frequency of 38 Hz. First microtubules are images both fixed and in live cell to validate the method and assess the resolutions. Then budding yeast Cdc12_GFP septins is assayed in vivo.

I am not an optical imaging expert but the methodology seems strong enough at least analyzing microtubules. Indeed as compared with classical SIM, fFE-SIM obviously provides better temporal and lateral resolution.

In the introduction, it is mentioned that SIM combined with SRFF achieves 32 nm resolution while the resolution obtained here is 80 nm.

However, I have strong concerns about the section dedicated to septins:

Within the section dedicated to septins the most interesting insight concerns the dynamic behavior of septins in between the two more stable fluorescent patterns. However, the weaker part concerns the description of septin ultrastructure.

The 80 nm lateral resolution is not sufficient to fully analyze the ultrastructural organization of septins. Indeed, septins filaments display a periodicity of about 30 nm and at 80 nm most of the details would be thereby discarded. Hence septins might not have been the best protein to investigate.

Also, do the authors know whether the septin signal they are visualizing correspond to endogeneous septins? Is there any over-expression of the protein using this yeast strain? Can they check for this?

In addition, as a control, the analysis of another septin protein that should in principle co localize with Cdc12 would be valuable.

The proposed model does not rely on the observed data and appears fully extrapolated. The data has been strongly over interpretated. It is a shame because it eventually weakens the main message which supports the emergence of this new imaging method. Hence, I would recommend removing Figure 4 and be more realistic with a model that would reflect the data and would not suffer from overinterpretation.

Indeed, it is difficult to agree with this 3D model relying on 2D images.

Eventually, as compared with the current knowledge in the septin field, we do not learn much more unless the resolution should be improved or the method should allow 3D visualization and/or reconstruction of the raw data.

I also have minor concerns:

The introduction appears strongly opinionated stating that most of the methodologies used before are worthless while they all brought extensive knowledge and progress, event being imperfect. For instance, the statement l50-51 is too strong suggesting that in vitro experiments “cannot represent the structure in vivo”. Likewise writing (L56-57): “recombination imaging and EM…do not reveal the the transformation process…” is too strong given that those methods have proben to be quite insightful.  Hence, I would thus recommend rewriting this section, changing the tone.

L36: « pol-ymerized » : The word should be split after the o or the y.

L38-39: The sentence should be rephrased, especially the “and its results” to be replaced maybe by “its study”.

L46: I would say “not fully known” instead of “unknown” given that people in the community do have some insights even though everything has not been deciphered.

Author Response

Point 1: In the introduction, it is mentioned that SIM combined with SRFF achieves 32 nm resolution while the resolution obtained here is 80 nm.

Response 1: Thanks for the comment. Since the real experimental environment is much more complicated than the theoretical conditions, it’s not so easy to achieve the best resolution as we mentioned in the introduction. Actually, the optical properties of the labelled samples’ fluorescent groups greatly affect the achievable resolution. For instance, if the fluorescent protein is easy to get quenched, the image quality and spatial resolving capability of final results would be decreased due to the lack of information. The situation is even severer when it comes to the living cells because the cytotoxicity of living cells, the laser power and exposure time would greatly affect the cell states and then decrease the real spatial resolution. Hence, the resolution of living cell imaging is likely to be less than that of fixed cells. According to the results shown in Figure S1, fFE-SIM achieves the resolution of 80 nm comparing with the result of conventional SIM (180nm).

Point 2: The 80 nm lateral resolution is not sufficient to fully analyze the ultrastructural organization of septins. Indeed, septins filaments display a periodicity of about 30 nm and at 80 nm most of the details would be thereby discarded. Hence septins might not have been the best protein to investigate.

Response 2: Thanks for the comment. We agree the 80 nm lateral resolution is not sufficient to analyse the periodical structures of septins at single-molecular level. However, the key information we want to convey here is the capability of our method to obtain clear highly dynamic structures in the middle, which are blurred after SIM reconstruction and can’t be seen under the electron microscope or other super-resolution microscopy method for fixed cell. What we want to emphasize here is the unclear parts of septins become visible after using fFE-SIM. This result shows potential to apply our methods in the samples having highly dynamic parts in live cell.

Point 3: Also, do the authors know whether the septin signal they are visualizing correspond to endogeneous septins? Is there any over-expression of the protein using this yeast strain? Can they check for this?

Response 3: Thanks, this is an important suggestion. In general, overexpression may produce aggregations, but such phenomena often occur on long-lived, structurally stable proteins. For Cdc12 protein, it will be recruited to the division ring during yeast division, and septin will change structurally with the division process. As described in the introduction, the protein on the septin ring has been dynamically exchanged with the septin protein pool in the cytoplasm. If there is an overexpressed protein, it will also be in the free protein pool. We were unable to distinguish the changes in the localization of overexpressed cdc12 protein. The common method of endogenous protein labeling is immunofluorescence. However, immunofluorescence samples need a series of treatments on cells. For suspended dividing yeast cells with cell wall, such treatment often destroys the structure of septins.

Point 4: In addition, as a control, the analysis of another septin protein that should in principle colocalize with Cdc12 would be valuable.

Response 4: Thanks, this is an important suggestion. As the structure we found is highly dynamic, it is difficult to obtain the two-color super-resolution images simultaneously in living cells. We have taken efforts to study suitable equipment which can obtain the structural information of more than two septin proteins at the same time, but it may take years.

Point 5: The proposed model does not rely on the observed data and appears fully extrapolated. The data has been strongly over interpretated. It is a shame because it eventually weakens the main message which supports the emergence of this new imaging method. Hence, I would recommend removing Figure 4 and be more realistic with a model that would reflect the data and would not suffer from overinterpretation.

Indeed, it is difficult to agree with this 3D model relying on 2D images.

Response 5: Thanks for the comment. The 3D simulation diagram in Figure 4 adds an intermediate highly dynamic part to the previously published 3D simulation diagram of septin. During the imaging of Cdc12, we surprisingly found an unreported highly dynamic part in the middle of the septin ring, so we inferred that it was a non-hollow structure. Because septin is a ring structure, close to intermediate symmetrical structure, it is relatively easy to deduce from 2D structure to 3D structure. In fact, most of the published 3D simulation diagrams of septin derive from 2D diagrams[1, 2]. Due to the limitation of the resolution, we do not know the specific details in the middle of septin ring. In the 3D model diagram in Figure 4, we want to express the possible spatial structure of septin ring, that is, it is non-hollow. Our results show that this method could apply to dynamic microbial super-resolution imaging in the range of about 80 nm. And 3D imaging has relatively low temporal resolution and spatial resolution of the Z-axis[3]. If we use 3D imaging, it is difficult to get super-resolution dynamic images.

Point 6: The introduction appears strongly opinionated stating that most of the methodologies used before are worthless while they all brought extensive knowledge and progress, event being imperfect. For instance, the statement l50-51 is too strong suggesting that in vitro experiments “cannot represent the structure in vivo”. Likewise writing (L56-57): “recombination imaging and EM…do not reveal the the transformation process…” is too strong given that those methods have proben to be quite insightful.  Hence, I would thus recommend rewriting this section, changing the tone.

Response 6: Thanks for the comment. That's a good suggestion. We'll make changes the Lines 51-53 of the latest manuscript into “Although in vitro imaging has high accuracy and resolution, the reconstructed structures in vitro cannot completely represent the natural structure in vivo.”

Lines 56-58 will be changed to “Moreover, in vitro recombination imaging and EM imaging do not perfectly reveal the transformation process in the higher-order structures of septins during cell division.”

Point 7: L36: « pol-ymerized » : The word should be split after the o or the y.

Response 7: Thanks for the comment, but we didn't find the word split in the latest version of the manuscript.

Point 8: L38-39: The sentence should be rephrased, especially the “and its results” to be replaced maybe by “its study”.

Response 8: Thanks for the comment. That's a good suggestion. We'll make changes the Lines 38-41 of the latest manuscript into “Budding yeast is still a valuable model for the study of higher-order structures of septins, and its study may also apply to animal cells, helping to understand septin-related diseases.”

Point 9: L46: I would say “not fully known” instead of “unknown” given that people in the community do have some insights even though everything has not been deciphered.

Response 9: Thanks for the comment. That's a good suggestion. We'll make changes the Lines 45-46 of the latest manuscript into “The process by which the septin filament controls the separation of the cell membrane between the mother and daughter cells is still not fully known.”

References

  1. Vrabioiu, A.M. and T.J. Mitchison, Structural insights into yeast septin organization from polarized fluorescence microscopy. Nature, 2006. 443(7110): p. 466-469.
  2. Ong, K., et al., Architecture and dynamic remodelling of the septin cytoskeleton during the cell cycle. Nature communications, 2014. 5(1): p. 1-10.
  3. Huang, B., M. Bates, and X. Zhuang, Super-resolution fluorescence microscopy. Annual review of biochemistry, 2009. 78: p. 993-1016.

Round 2

Reviewer 2 Report

I still have an issue with Figure 4 and the corresponding conclusions.

The 2D data does not support this 3D model. Hence I still suggest to remove Figure 4 which is too speculative.

I agree that the main and most valuable results is the description of the dynamic organization of septins at the yeast bud neck.

I have an additional question: It seems that even at low resolution the septin signal overlap with the membrane significantly which differs from observations made by other labs (see Figure 3). Can the author explain and comment on this?

Author Response

Point 1: The 2D data does not support this 3D model. Hence I still suggest to remove Figure 4 which is too speculative.

Response 1: Thanks for the comment. That's a good suggestion. We'll delete Lines 273-309 of the latest manuscript and Figure S4.

We'll add a sentence on Line 262 “Video 4 shows dynamic organization of septins at the yeast bud neck in the several other cells.”

Point 2: I have an additional question: It seems that even at low resolution the septin signal overlap with the membrane significantly which differs from observations made by other labs (see Figure 3). Can the author explain and comment on this?

Response 1: Thanks for the comment. We did not fluorescent label the cell membrane. The membrane you are referring to may be the Cdc12 proteins that is not localized to the budding neck. Moreover, the AH domain of cdc12 can interact with the membrane, which is very close to the cell membrane.

Round 3

Reviewer 2 Report

The manuscript can now be published with the additional modifications.